# A Scenario-Based Case Study: Using AI to Analyze Casualties from Landslides in Chittagong Metropolitan Area, Bangladesh

**Edris Alam** [1,2,*] **, Fahim Sufi** [3,*] **and Abu Reza Md. Towfiqul Islam** [4]

1 Faculty of Resilience, Rabdan Academy, Abu Dhabi P.O. Box 114646, United Arab Emirates
2 Department of Geography and Environmental Studies, University of Chittagong, and Disaster Action and Development Organisation (DADO), Chittagong 4331, Bangladesh
3 School of Public Health and Preventive Medicine, Monash University, Melbourne, VIC 3000, Australia
4 Department of Disaster Management, Begum Rokeya University, Rangpur 5400, Bangladesh
* Correspondence: ealam@ra.ac.ae (E.A.); research@fahimsufi.com (F.S.)

**Abstract:** Understanding the complex dynamics of landslides is crucial for disaster planners to make timely and effective decisions that save lives and reduce the economic impact on society. Using the landslide inventory of the Chittagong Metropolitan Area (CMA), we have created a new artificial intelligence (AI)-based insight system for the town planners and senior disaster recovery strategists of Chittagong, Bangladesh. Our system generates dynamic AI-based insights for a range of complex scenarios created from 7 different landslide feature attributes. The users of our system can select a particular kind of scenario out of the exhaustive list of $1.054 \times 10^{41}$ possible scenario sets, and our AI-based system will immediately predict how many casualties are likely to occur based on the selected kind of scenario. Moreover, an AI-based system shows how landslide attributes (e.g., rainfall, area of mass, elevation, etc.) correlate with landslide casualty by drawing detailed trend lines by performing both linear and logistic regressions. According to the literature and the best of our knowledge, our CMA scenario-based AI insight system is the first of its kind, providing the most comprehensive understanding of landslide scenarios and associated deaths and damages in the CMA. The system was deployed on a wide range of platforms including Android, iOS, and Windows systems so that it could be easily adapted for strategic disaster planners. The deployed solutions were handed down to 12 landslide strategists and disaster planners for evaluations, whereby 91.67% of users found the solution easy to use, effective, and self-explanatory while using it via mobile.

**Keywords:** AI; landslides; causalities; hazards

## 1. Introduction

Landslides are natural phenomena that have an adverse effect on human life, as well as the economy [1]. For the purpose of reducing the negative impact of landslides and to have an increased level of disaster preparedness [2], it is crucial to have a multi-dimensional understanding the attributes of landslides. The complex nature of landslide dynamics makes it extremely difficult to understand the impact of a particular type of landslide. Bangladesh is susceptible to a variety of natural and human-induced hazards including tropical cyclones, floods, droughts, earthquakes, tsunamis, and landslides [2]. In particular, landslides have become recurrent phenomena in the Southeast Bangladesh in recent decades. Therefore, the Government of Bangladesh (GoB) and its coastal residents have been engaged in reducing resultant deaths from tropical cyclones, but landslides have still caused over 500 deaths in Southeast Bangladesh with the majority occurring in informal settlements in Chittagong and Rangamati districts since 2000. The root causes contributing to the vulnerability of three different communities in the southeast part of Bangladesh. These communities are Bengali, Tribal, and Rohingya refugees, [3] and effective local risk governance was also promulgated [4]. Studies were also conducted to identify the root causes and impacts of landslides using qualitative methods (e.g.,

interviews and surveys) in Chittagong city and Rangamati district [5]. However, there is further scope for applying artificial intelligence (AI)-driven techniques to identify physical parameters that significantly influence deaths associated with landslides. As such, in this paper, we deployed a new scenario-based AI insight system, that facilitates an in-depth understanding of landslide hazards, enhances "risk perception", and raises the level of "disaster preparedness" in relation to landslides.

Geo-structural and causative factor-based analyses were applied for exploring landslide susceptibility zoning. Landslide susceptibility and risk assessment have been studied at global levels [6,7]. Geo-spatial technologies such as the application of geographical information systems (GIS), global positioning systems (GPS), and remote sensing (RS) have recently taken prominence in hazard assessment and risk identification to assist in decision making related to landslide disaster risk management [8,9]. GPS is a space-based navigation satellite system that acquires information relating to exact location and time in all weather conditions, anywhere in the world and it assists with the collection and storage of landslide information. GIS is used in collecting, storing, and analyzing geographic information and their non-spatial attributes. A plethora of studies have been conducted using GIS for landslide hazard and risk assessment [10]. Remote sensing is a system where information about the earth's surface is obtained without direct contact with it. In recent decades, RS has been widely applied for the identification of landslide areas, vulnerability, and risk mapping [11]. Apart from the aforementioned techniques, machine learning algorithms are gaining prominence in enhancing disaster preparedness and response.

There are varieties of methods available to study landslide susceptibility. These include but are not limited to landslide inventory-based probabilistic, deterministic, heuristic, and statistical techniques [12]. The most used landslide inventory-based probabilistic techniques involve the development of the inventory of landslides, geo-morphological analysis, and generating susceptibility maps based on provided parameters [13]. Deterministic approaches are also familiar as quantitative methods that involve quantifying factors such as physical factors, e.g., soil, rainfall, vegetation, and slope variables to generate maps that display the spatial distribution of input data [14,15]. A qualitative approach (heuristic analysis) involves analyzing aerial photographs or conducting field surveys to identify the intrinsic properties of a landform [16]. Statistical analysis uses sample data to identify the relationship between the dependent variable (the presence or absence of landslides), and the independent variables (landslides triggering/causative factors [17].

Artificial intelligence (AI) methods use some of the statistical concepts. These methods are based on assumptions, predetermined algorithms, and output. AI methods or machine learning methods that are used for landslide studies include artificial neural networks (ANN), fuzzy-based, hybrid, kernel-based, and tree-based methods [18]. These methods are suitable for generating results regardless of data types (i.e., both discrete and continuous data) and data limitation (i.e., the types and number of conditioning factors). For example, the research in [19] uses machine learning algorithms to understand the complex dynamics of global landslides which may help strategic decision makers.

Although these studies provide valuable insight into landslide susceptibility as well as the causes and impacts of landslides on the poor in Chittagong, there is a dearth of research that focuses on using AI systems to analyze casualties from landslides on a small scale. Reducing disaster deaths through AI at both the national and local levels is aligned with the United Nations' Sendai Framework (2015–2030) for Disaster Risk Global Target A: 'Substantially reduce global disaster mortality by 2030' and Global Target G: 'Substantially increase the availability of and access to multi-hazard early warning systems and disaster risk information and assessments to people by 2030 [20]. Since Bangladesh is a signatory to the Sendai Framework; it is important that multi-hazard early warning systems and disaster risk information for all hazards are available at the community level by the year 2030.

In this paper, first, we designed and developed a new scenario-based AI insight system that can connect to a landslide database, so as to find out unknown insights from

landslide data. Second, we connected our scenario-based AI insight system to a dataset containing landslide information and finally, we demonstrated the dynamic generation of AI-based insights based on specific scenarios. It should be noted that the methodology described within this paper allows for the automatic generation of AI insights, without the need to manually execute statistical methods. As opposed to the traditional statistical methods, where a data scientist is required to manually prepare, execute, and analyze, the methodology presented in this paper automates the entire process and provides AI-driven insights in a fast and efficient manner. The results in Section 3 (results) show the positive correlation of area of mass as well as rainfall towards the number of casualties.

Equipped with these AI insights, a disaster recovery planner and strategist can make informed, timely, and evidence-based decisions that can save lives and reduce the economic impact of likely disasters on a society. Moreover, the AI insights would support policy planners in understanding the characteristics of landslides in a particular area and provide useful guidance for policy implementation.

## 2. Materials and Methods

First, the data was obtained from previous landslide catalogues, local histories, archive of institutional and administrative records, newspapers, reports, digital archives, and published peer-reviewed journal papers dedicated to landslides in the Chittagong Metropolitan Area (CMA), and subsequently cleaned and transformed before modelling. Data collected from secondary sources were validated through field visits and investigations to identify accurate locations of landslide occurrences. Following this, data modelling using the best practice was performed and then the data was visualized and analyzed using AI systems and algorithms. The details of these AI-based analyses are portrayed within this section. Finally, data-driven insights were generated. Figure 1 demonstrates the step-by-step process for generating AI insights on the CMA landslide data. The following subsections describe the study area selection, the sources of the data, preparation of the data, modelling of the data, visualization of the data, and analysis with AI-based algorithms (like linear regression, logistic regression, and decomposition tree analysis).

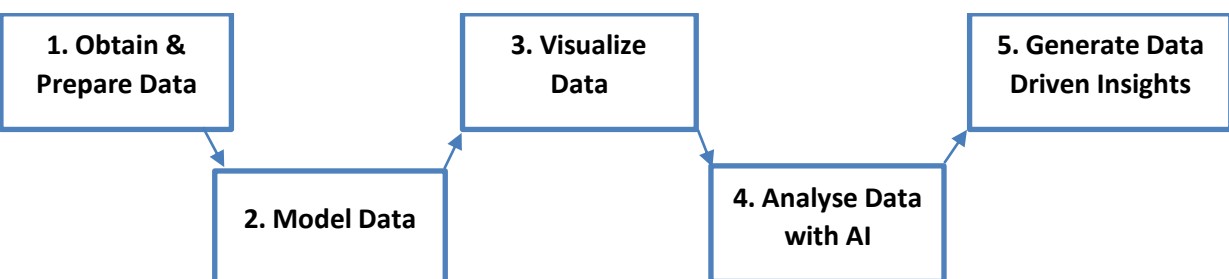

**Figure 1.** High level methodology of AI insight system for analyzing landslides in the CMA of SE Bangladesh, particularly the landslide susceptible areas in the Chittagong, Rangamati, and Cox's Bazar districts.

### 2.1. Study Area Selection

Chittagong was selected as the area of study, as the residents of the city have experienced a record of deaths associated with landslides since the year 2000. For example, on the 11 June 2007, landslide events alone caused the death of 128 casualties and 100 injuries in places adjacent to hilly areas, because landslides were triggered by heavy rainfall (610 mm) for eight consecutive days. Five years later, on the 26 June 2012, another eight days of continuous rainfall (889 mm) triggered landslides that led to 90 casualties [2]. These landslide events occurred in hill cutting areas that are characterized by high angles/slopes. Slope failure in these fragile hilly areas occurs during the rainy season between the months of June and September. It is important to note that the population in Chittagong

has increased six times in number since 1974, creating a significant number of people that live in highly vulnerable areas (Figure 2).

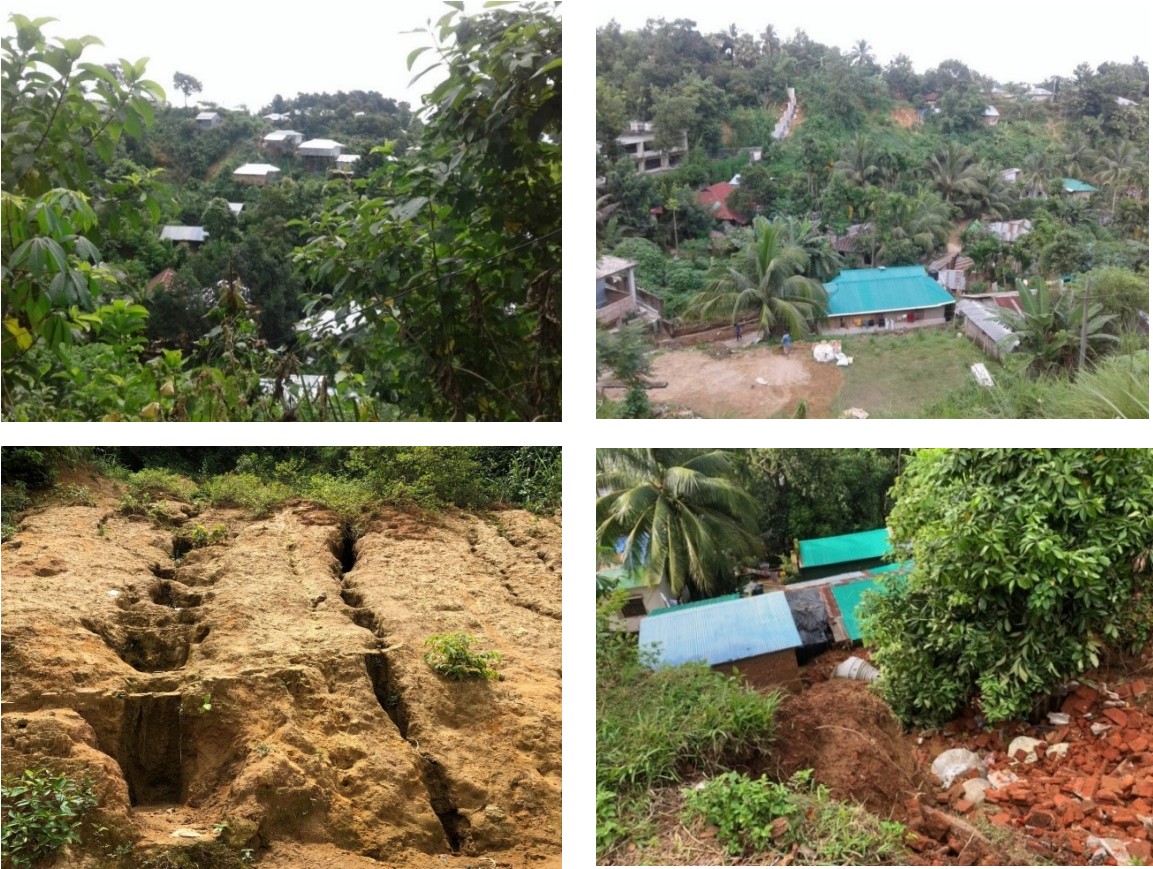

**Figure 2.** Landslide vulnerability in different areas of Chittagong (map and photo) (Source: Field visit, October 2018).

Chittagong lies along the western margin of the tectonically active Chittagong-Tripura Fold Belt. The district is located between 20′35°N and 22′59° N latitude, and 91′27° E and 92′22° E longitude (Figure 3). Hills in the district are mainly composed of weathered and loose sedimentary rocks of tertiary (65–1.8 Ma) age, which are prone to landslides. The mean monthly maximum and minimum temperatures range between 78.76–90.44° F and 55.88–77.38° F, and the monthly average minimum rainfall is 0.66 mm in the month of January and maximum rainfall 74.70 mm in the month of July. The average rainfall per year is about 2794 mm [21]. The northwestern and monsoon clouds are primarily responsible for the rainfall in the area and almost 90% of the total yearly precipitation takes place between the months of June and October [21]. The total area of Chittagong City Corporation is about 170.41 km². The urban population of the Chittagong district was only 0.90 million in 1974 which increased to 5.13 million in 2021, representing an approximate increase of six times of the urban population in the last 47 years [21].

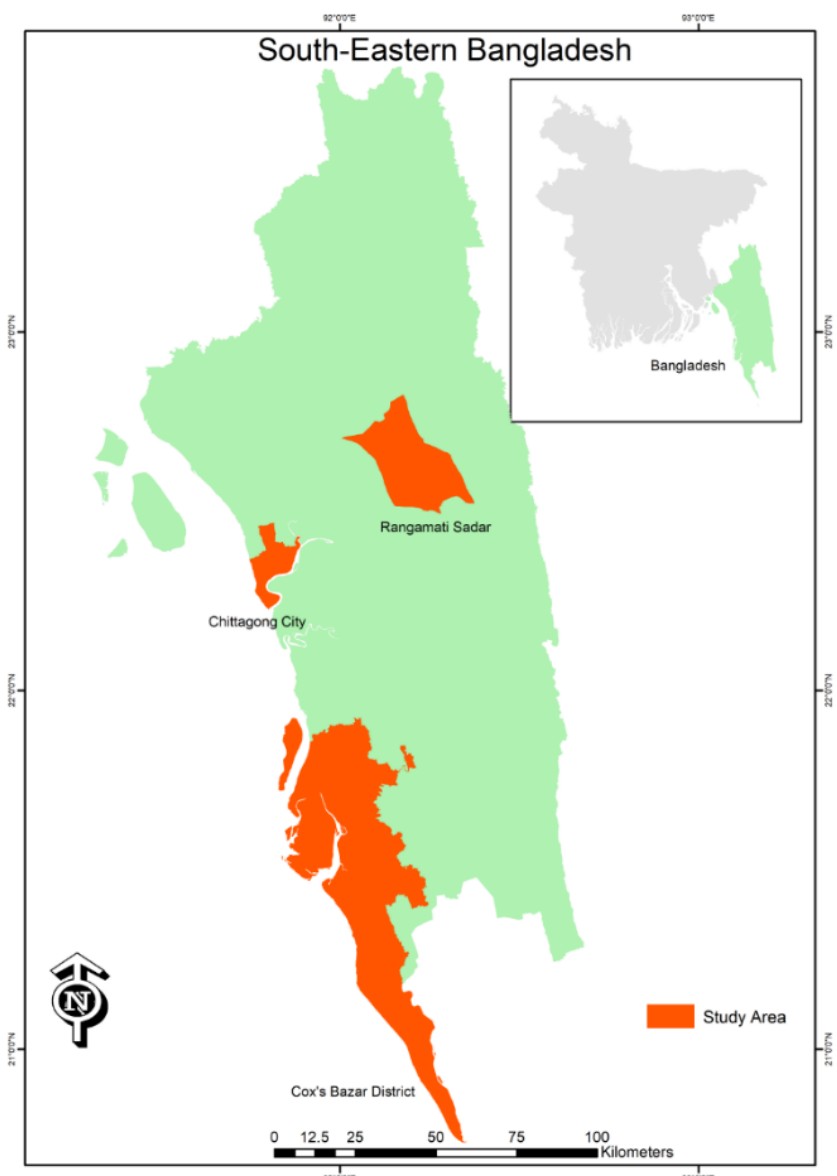

**Figure 3.** Location of SE Bangladesh, particularly the landslide susceptible areas in the Chittagong, Rangamati, and Cox's Bazar districts.

### 2.2. Obtaining and Preparing the Data

Data can be sourced from one or more sources and these sources can be multiple, ranging from online databases, websites, excel files, flat files, web-based application programming interfaces (APIs), or even PDF files. After identifying the data sources, the data was obtained with integration tools like SQL Server Integration Services (SSIS), Power BI Query Editor, Oracle Data Integrator, Tibco Pervasive Integration, etc. These data integration tools facilitate the export, transform, load (ETL) process, which obtains data from many different sources and forms them into a data warehouse, whereas specialized programming languages like the Mashup (M) language is used for data transformations and data cleansing.

Data transformation and data cleansing can be referred to as "data preparation", since data needs to first be transformed into the right format before the data is modelled or analyzed. For our research, we obtained publicly available data directly from PDF files [22] and then we transformed the data into a suitable format that allowed for faster analysis. Following data transformation, the feature attributes of the CMA landslide data can be better understood after completing data preparation. Table 1 shows the detailed statistics

of the CMA landslide data. Understanding the statistics for the CMA landslide feature attribute details is crucial before proceeding to the next steps of the methodology, namely modelling the data, visualizing the data, and analyzing the data with AI.

**Table 1.** Landslide attribute, data type, and data distribution.

| Type of Attribute | Data Type | Attribute Distribution | Other Attribute Details |
|---|---|---|---|
| ID | Integer | 57 distinct, 57 unique | 57 Distinct, 57 Unique Value Example: Ranges from 1 to 57 |
| Latitude | Decimal | 50 distinct, 44 unique | 50 Distinct, 44 Unique |
| Longitude | Decimal | 54 distinct, 51 unique | 54 Distinct, 51 Unique |
| Elevation | Decimal | 56 distinct, 55 unique | 56 Distinct, 55 Unique |
| Date | Date (dd-mm-yyyy) | 6 distinct, 0 unique | 6 Distinct, 0 Unique, 34 Empty<br>● Valid 40%<br>● Error 0%<br>● Empty 60% |
| Hill Name | Text | 29 distinct, 13 unique | 29 Distinct, 13 Unique Value Example: Lebu Bagan, Ctg. University, Foy'z Lake Zoo Hill, Medical Hill, Tankir Pahar, Sekandar Para, etc. |
| Area of Mass | Decimal | 56 distinct, 55 unique | 56 Distinct, 55 Unique |
| Types | Text | 3 distinct, 0 unique | 3 Distinct, 0 Unique Value Example: Slide, Fall, Topple |
| State | Text | 4 distinct, 0 unique | 4 Distinct, 0 Unique Value Example: Active, Stabilized, Dormant, Reactivated |
| Style | Text | 2 distinct, 0 unique | 2 Distinct, 0 Unique Value Example: Single, Successive |

**Table 1.** *Cont.*

| Type of Attribute | Data Type | Attribute Distribution | Other Attribute Details |
|---|---|---|---|
| Rainfall | Integer | 10 distinct, 4 unique | 10 Distinct, 4 Unique, 18 Empty<br>● Valid    68%<br>● Error    0%<br>● Empty    32% |
| Casualty | Integer | 12 distinct, 8 unique | 12 Distinct, 8 Unique |

### 2.3. Modelling the Data

Data modelling is the most important stage in the process of generating data-driven insights and when it is done correctly, an AI-driven solution can produce powerful insights with minimum delay. During this stage, relationships among different sets of data are drawn with the right cardinality.

As seen in Figure 4, the data obtained for this paper were arranged in a star schema [23], where the main factual data resides in the center (referred to as landslide DB). Surrounding the fact tables, there are dimension tables that include: types, state, date, hill name, and style. This arrangement of star schema allows for the control of the fact table (i.e., Landslide DB) with one-way filtering of information by type, state, date, and hill name as well as style. The main benefit of the star schema technique over other data modelling techniques (e.g., flattened table, snowflake, etc.) is the speed, since it provides more accurate results during data analysis [24].

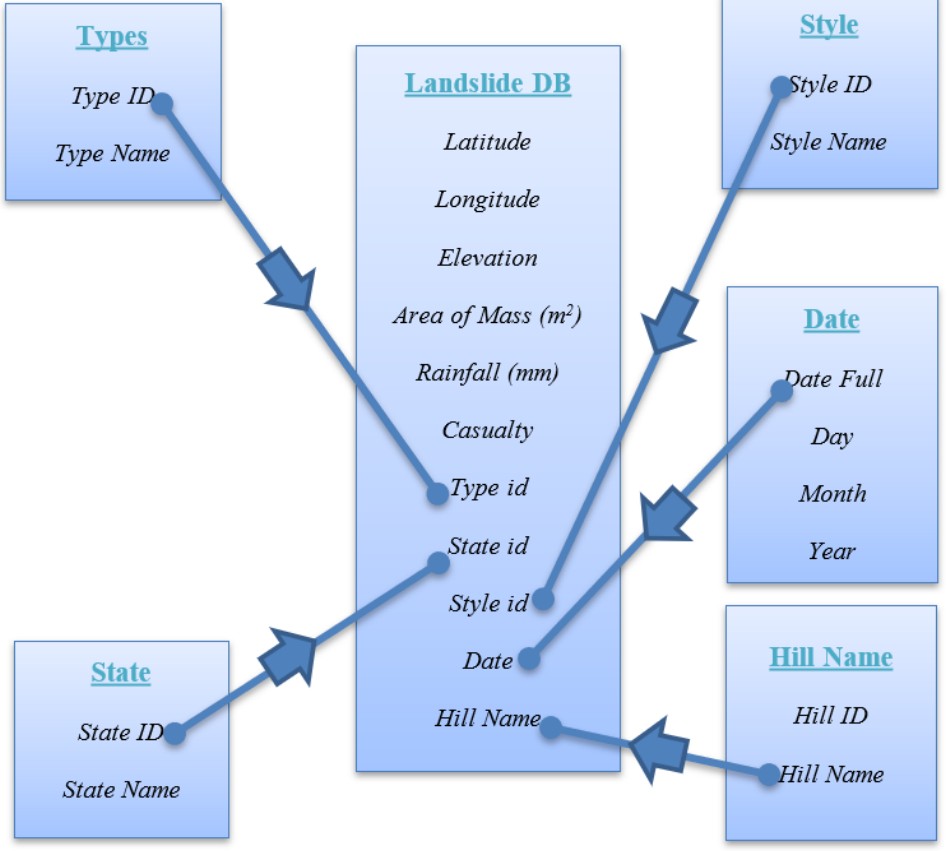

**Figure 4.** Data modelling of the CMA landslide database.

### 2.4. Visualizing the Data

Once the data modelling was completed, we used state, rainfall (mm), elevation (m), and type information to filter the factual data that drives the AI-based insights. A wide range of visualizations like slicer, Bing Maps, and key influencers were used. Changing the values for each of the filters (e.g., state to dormant or stabilized), filters the fact table landslide DB, which in turn changes the key influences (Figure 5).

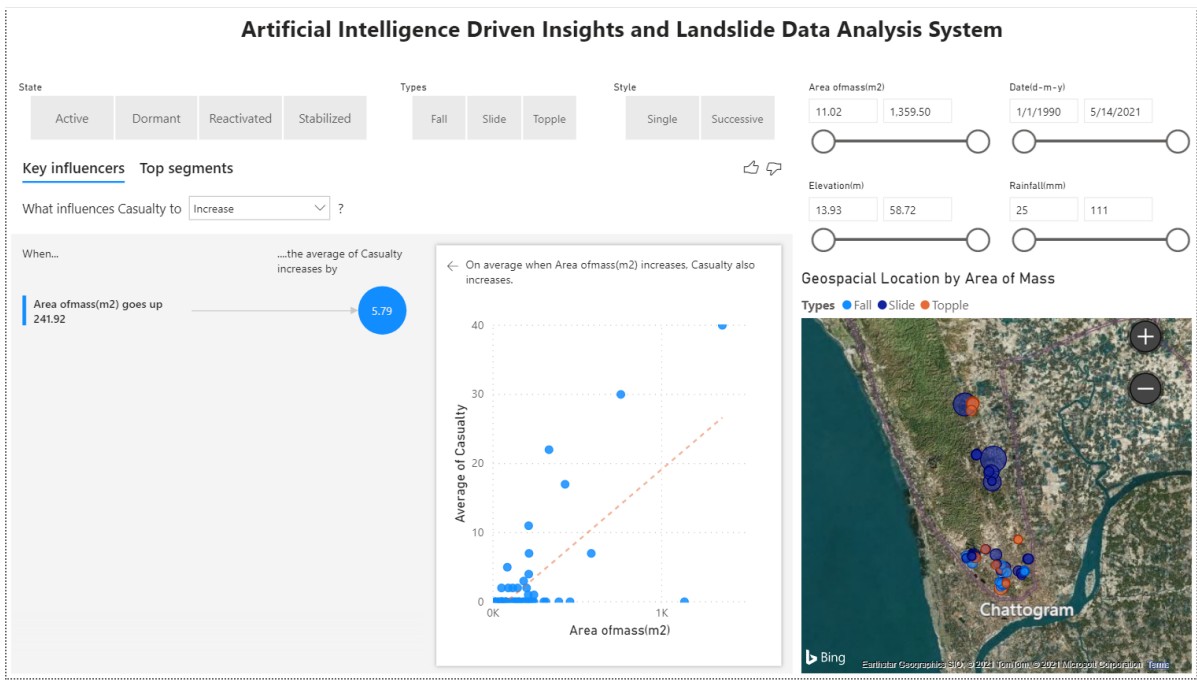

**Figure 5.** AI-based insights and landslide analysis system.

### 2.5. Analyzing Data with AI

This paper focused on automatically identifying the relationships that may exist between an outcome variable (i.e., landslide related casualty) with a range of other variables (e.g., rainfall, area of mass, elevation, etc.). Therefore, we used a particular AI-based regression tool called the key influencers visualization (i.e., https://learn.microsoft.com/en-us/power-bi/visuals/power-bi-visualization-influencers?tabs=powerbi-desktop, accessed on 20 December 2022). There are many other AI-based as well as non-AI-based statistical techniques that may suit other research objectives. For example, to find the similarities and dissimilarities between past landslides, AI-based automated clustering techniques could be used. Within this research, Microsoft Power BI's Key Influencer visualization was used to analyze casualties (from landslide) and they were explained by the following list of feature attributes as named below:

- Area of Mass (m$^2$)
- Elevation (m)
- Hill Name
- Rain fall (mm)
- State
- Style
- Types
- Date

This analysis used machine learning algorithms provided by ML.NET [25] to figure out what matters the most in driving landslide feature attributes. As seen in Figure 6, the analysis process uses the CMA landslide data, ranks the factors that matter, contrasts

the relative importance of these factors, and displays them as key influencers for both categorical and numeric metrics.

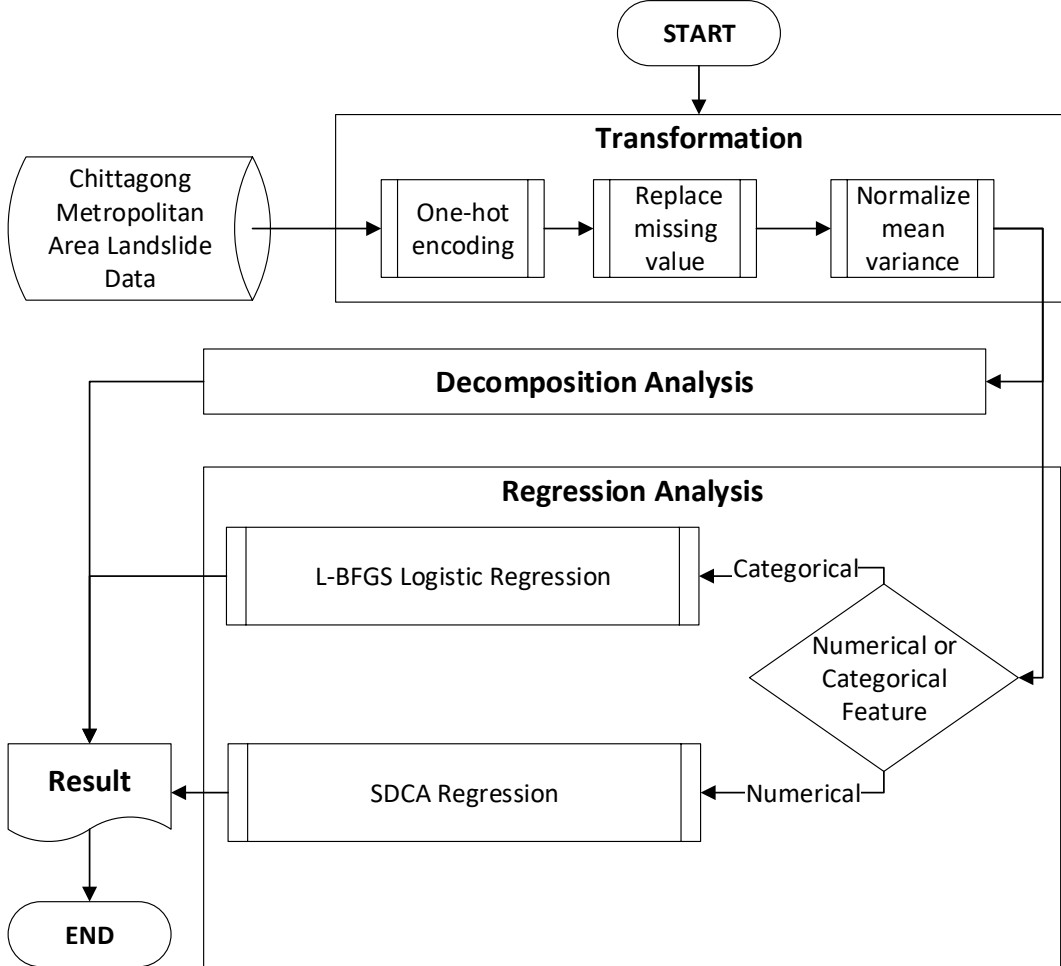

**Figure 6.** The process of obtaining AI insights from CMA landslide data using machine learning algorithms.

As seen in Figure 6, two main categories of AI-based statistical analysis are executed on the CMA Landslide data, namely transformation [25], decompression analysis [25], and regression analysis. Transformation analysis is executed for preparing the CMA landslide data before running the regression analysis. Within the transformation, three algorithms are executed and they include:

One-hot encoding: Calling on the OneHotEncoding() method within Microsoft.ML.Transforms class results in a conversion of categorical information into numeric values for efficient and effective processing of machine learning algorithms [25].

Replacing missing value: Calling on the ReplaceMissingValues() method within Microsoft.ML.Transforms class results in a replacement of the missing value with either default, minimum, maximum, mean, or the most frequent value [25].

Normalize mean variance: Calling on the NormalizeMeanVariance() method within Microsoft.ML.Transforms class results in an adjustment of values measured on different scales to a notionally common scale with computed mean and variance of the data [25].

Once the CMA landslide data are prepared for regression analysis, two different types of regressions are performed. For numerical features, linear regression is performed using Microsoft's ML.Net's SDCA regression implementation [19,25]. Linear regression is one of the simplest machine learning algorithms that falls under supervised learning techniques, and it is used for solving regression problems. Moreover, it is used for predicting the

continuous dependent variable with the help of independent variables. The goal of the linear regression is to find the best fit line that can accurately predict the output for the continuous dependent variable by finding the best fit line, so that the algorithm establishes the linear relationship between dependent variable and independent variable in the form of $y = b_0 + b_1 x_1 + \varepsilon$. On the other hand, for the categorical feature, logistic regression is performed using ML.Net's L-BFGS logistic regression [26]. Logistic regression is one of the most popular machine learning algorithms that falls under supervised learning techniques since it can be used for classification as well as for regression problems. Logistic regression is used to predict the categorical dependent variable with the help of independent variables using $\log\left[\frac{y}{1-y}\right] = b_0 + b_1 x_1 + b_2 x_2 + \ldots b_n x_n$. As seen in Figure 6, depending on the variable type (i.e., categorical or numerical), either logistic or linear regression is selected.

Other than using linear regression and logistic regression, this study also used decomposition analysis with a decomposition tree. Decomposition tree visualization is a valuable tool for ad hoc exploration and for conducting root cause analysis, whilst allowing the user to visualize the data across multiple filter attributes or dimensions.

Our implementation of decomposition analysis allows for the visualization of landslide casualty data over a range of landslide feature attributes, namely: area of mass, elevation, rainfall, state, and types. As shown in Figure 7, interactive root cause analysis and data exploration were supported by the aggregation of data and drill-down, where a user can click and find out what feature attribute causes the highest or lowest number of casualties.

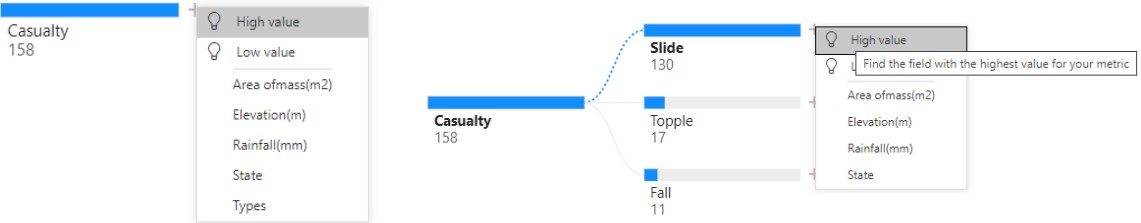

**Figure 7.** Decomposition tree visualization allows the user to perform interactive analysis by area of mass, elevation, rainfall, state, and types.

For feature attributes (i.e., area of mass, elevation, rainfall, state, type, date, etc.), $T = \{T^1, T^2, T^3, \ldots, T^N\}$, where $N$ is the number of total filter attributes within a dataset (i.e., the cardinality of T, $|T| = N$), each feature attribute can form one or many filtered conditions, as follows:

$$T^1 = \left\{T^1_1, T^1_2, T^1_3, \ldots, T^1_P\right), \text{ such that } \left|T^1\right| = P \qquad (1)$$

$$T^2 = \left\{T^2_1, T^2_2, T^2_3, \ldots, T^2_Q\right), \text{ such that } \left|T^2\right| = Q \qquad (2)$$

$$T^3 = \left\{T^3_1, T^3_2, T^3_3, \ldots, T^3_U\right), \text{ such that } \left|T^3\right| = U \qquad (3)$$

$$T^N = \left\{T^N_1, T^N_2, T^N_3, \ldots, T^3_N\right), \text{ such that } \left|T^3\right| = V \qquad (4)$$

Each of these filter conditions can filter r number of rows, $r \in \{1, 2, 3, \ldots R\}$ from the dataset. Proceeding with this context, we defined casualty count from landslides as Equation (5).

$$C^n_i = \sum_{i=0}^{r}(casulty\_count), \text{ Where, } r \text{ is the rows effected by filter attribute condition } T^n_i \qquad (5)$$

Our decomposition tree visualization (supported by AI) allows the user to find the next filter attribute condition to drill down into, based on either high or low values.

High Value: This mode considers all available filter attribute conditions and determines which one to drill into to obtain the highest value of the measure being analyzed. Therefore, the high-value AI split mode finds the most influential filter attribute condition $T_i^n$, for which the highest level of casualties occur, which is represented by

$$\exists T_i^n \subseteq T \mid C_i^n > C_j^m, \ \forall n, m \subseteq \{1, 2, 3, \ldots, N\} \land \forall i, j \subseteq \{1, 2, 3, \ldots\} \tag{6}$$

Low Value: This mode considers all available filter attribute conditions and determines which one to drill into to obtain the lowest value of the measure being analyzed. Therefore, the low-value AI split mode finds the most influential filter attribute condition $T_i^n$, for which the lowest level of casualties occur, which is represented by

$$\exists T_i^n \subseteq T \mid C_i^n < C_j^m, \ \forall n, m \subseteq \{1, 2, 3, \ldots, N\} \land \forall i, j \subseteq \{1, 2, 3, \ldots\} \tag{7}$$

In this way, the AI split allows the user to understand the details of the root cause. This AI split-based decomposition analysis was used in our most recent study on knowledge discovery for landslides globally.

### 2.6. Generating Data-Driven Insights

In this phase, valuable insights are produced. The success of this phase depends on the success of previous activities such as preparation (i.e., transformation and cleaning) of the data, selection of the right AI visualization, and most importantly data modelling. Following the employment of the AI-based key influencers visualization, the key factor that influences the number of casualties was area of mass (m$^2$). The other factor that influences the number of casualties under specific conditions was elevation (m). Data-driven insights are generated by configuring one or more scenarios. A scenario can easily be created using our system either by clicking the desired buttons shown in Figure 8 or by changing the sliders as shown in Figure 8. In our system, we have created scenarios (S) from 7 different attributes namely, types ($T^1$), state ($T^2$), style ($T^3$), elevation ($T^4$), area of mass ($T^5$), rainfall ($T^6$), and date ($T^7$) as parameters. Therefore,

$$S = [x, y, z, m, n, p, q \mid x \subseteq T^1, y \subseteq T^2, z \subseteq T^3, m \subseteq T^4, n \subseteq T^5, p \subseteq T^6, q \subseteq T^7] \tag{8}$$

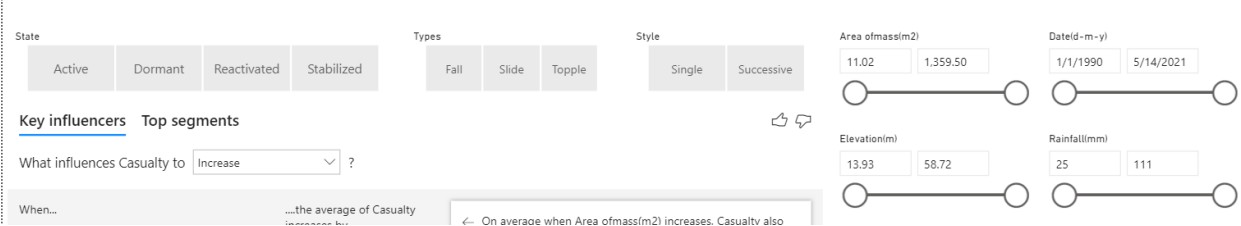

**Figure 8.** Filter area for the selection of landslide attributes.

Equation (9) to Equation (15) provide the unique values obtained from our dataset for each of the landslide parameters. For example, as seen from Equation (9), possible values for types ($T^1$) were "slide", "fall", or "topple". Similarly, the possible values for state ($T^2$) were "active", "stabilized", "dormant", or "reactivated" as seen from Equation (10). On the other hand, style ($T^3$) had only two unique values (i.e., single and successive) as it appears from Equation (11).

$$T^1 = \{Slide, Fall, Topple\} \tag{9}$$

$$T^2 = \{Active, Stabilized, Dormant, Reactivated\} \tag{10}$$

$$T^3 = \{Single, Successive\} \tag{11}$$

$$T^4 = \{ 13.93, 15.11, 15.93, 18.1, 18.1, 19.33, 19.84, 21.31, 21.59, 22.64, 23.12,$$
$$23.5, 24.71, 26.57, 26.98, 27, 28.41, 29.28, 30.82, 31.66, 32.39, 32.44, 32.56, 34.21,$$
$$34.63, 35, 35.18, 36.68, 37.54, 37.64, 37.92, 38.51, 38.64, 39.81, 40.19, 40.68,$$
$$41.18, 41.22, 44.26, 44.46, 45.12, 45.36, 45.42, 45.69, 46.07, 46.4, 46.51, 47.04,$$
$$48.36, 48.51, 48.67, 50.12, 51.79, 55.03, 55.95, 56.36, 58.72\} \tag{12}$$

$$T^5 = \{ 11.02, 15.03, 16.5, 31.67, 33, 45.86, 47.04, 50.17, 50.26, 52.3, 56.05,$$
$$59.1, 71.93, 71.93, 75.88, 76.43, 77.81, 84.56, 89.91, 105.38, 116.32, 118.34, 126.7,$$
$$130.32, 136, 145.06, 145.5, 152.79, 153.55, 157.07, 175.81, 181.7, 184.13, 188.59,$$
$$191.64, 198.89, 208.57, 209.12, 211.06, 211.61, 212.7, 213.26, 226.23, 232.52,$$
$$233.06, 241.79, 242.53, 301.06, 313.42, 331.84, 390.34, 427.04, 456.7,$$
$$582.27, 757.61, 1134.77, 1359.5\} \tag{13}$$

$$T^6 = \{\varnothing, 25, 26, 46, 50, 54, 55, 77, 88, 111\} \tag{14}$$

$$T^7 = \{\varnothing, 11/6/2007, 1/1/1990, 1/7/2011, 3/8/2005, 5/14/2021\} \tag{15}$$

Equations (14) and (15) contains null values represented by $\varnothing$.

To calculate the number of possible scenarios, we first need to calculate the possible filter options for each of the feature attributes. For example, as it appears from Equation (8), the type attribute could have the following filter options:

{}
{*Fall*}
{*Topple*}
{*Slide*, *Fall*}
{*Fall*, *Topple*}
{*Slide*, *Topple*}
{*Slide*, *Fall*, *Topple*}

Therefore, for the type attribute, there could be 7 possible filter settings as represented by $(2^{|T^1|} - 1)$, and the formula to calculate a power set of type attribute minus 1 (i.e., $P(T^1) - 1$). The number 1 is deducted since the power set also includes empty set and the selection of empty set is not a supported option by the system presented.

Hence, the total number of possible scenarios can be calculated as,

$$|S| = (2^{|T^1|} - 1) \times (2^{|T^2|} - 1) \times (2^{|T^3|} - 1) \times (2^{|T^4|} - 1) \times (2^{|T^5|} - 1) \times (2^{|T^6|} - 1)$$
$$\times (2^{|T^7|} - 1) = 1.054 \times 10^{41} \tag{16}$$

The purpose of this section is not only to produce an exhaustive list of insights from the landslide data, but also to demonstrate the ability of the designed AI solution for producing insights on any scenario out of the $1.054 \times 10^{41}$ possible scenarios (as shown in Equation (16)). In the next section we will explore results (i.e., AI insights obtained from a few of these scenarios).

## 3. Results

This study was reported based on two different AI-based techniques namely: automated regression analysis and decomposition analysis. Therefore, within the results section we will briefly describe AI insights derived from both methodologies. Table 2 demonstrates the outcome of conducting regression analysis on various scenarios. Seven rows of Table 2 represent five different scenarios since row 3 and row 4 represent the same scenario and row 5 and row 6 represent another single scenario. Table 2 has three columns representing the AI-based insight, the results obtained through the system interface, and the scenario condition. Row 5 and row 7 of Table 2 has the following scenario condition (i.e., both belong to the same scenario):

1.   State = "*Stabilized*",
2.   Type = *All*,
3.   Style = *All*,

4. Area of Mass = *All*,
5. Date = *All*,
6. Elevation = *{p | 29.05 ≤ p ≤ 58.72}*,
7. Rainfall = *{n | 43 ≤ n ≤ 111}*

**Table 2.** Ai insights generated on specific scenarios.

| AI Insight | AI-Based System Settings | Scenario |
|---|---|---|
| 1. When area of mass goes up 241.92, the average of causalities increases by 5.79 | 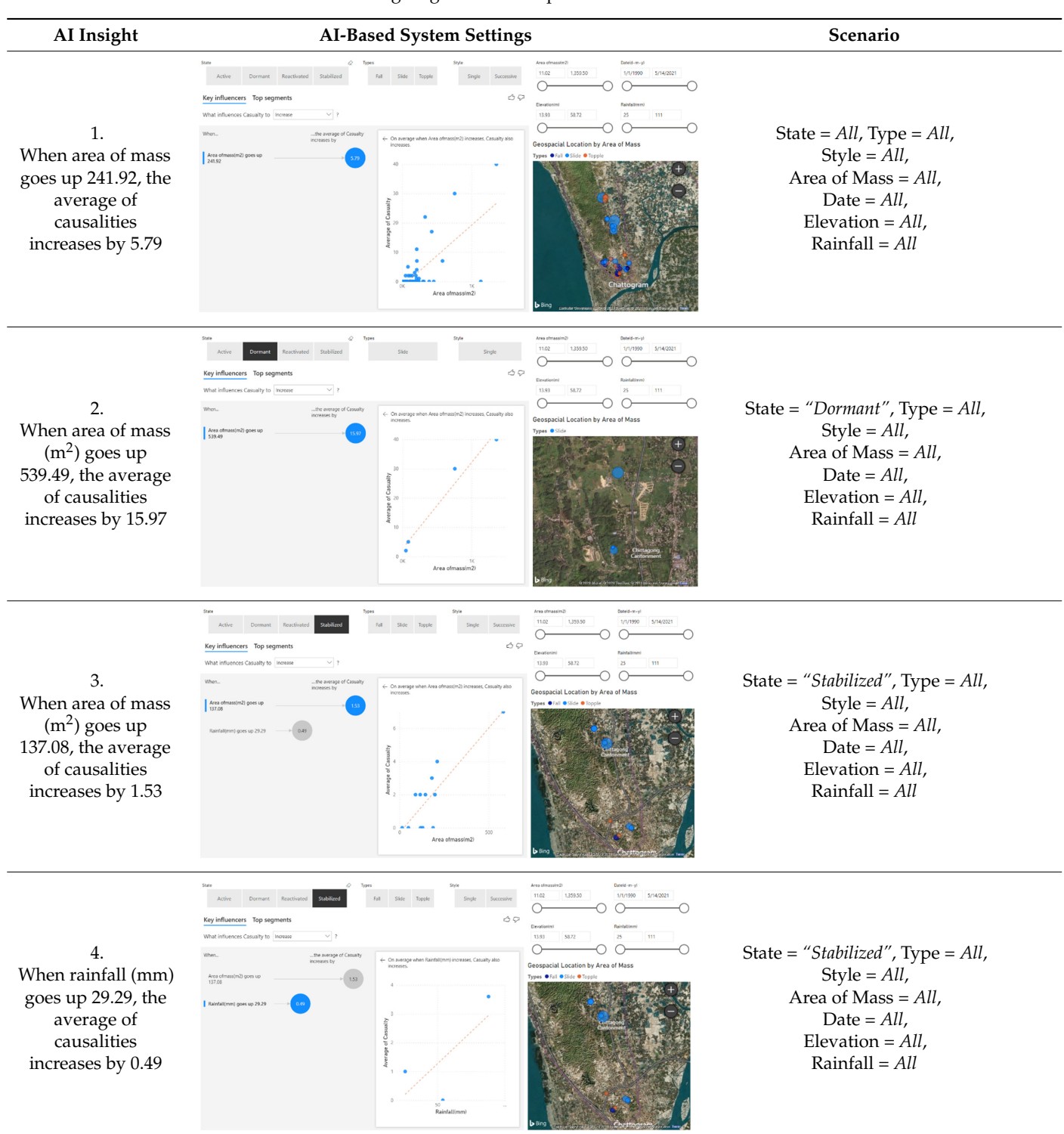 | State = *All*, Type = *All*, Style = *All*, Area of Mass = *All*, Date = *All*, Elevation = *All*, Rainfall = *All* |
| 2. When area of mass (m²) goes up 539.49, the average of causalities increases by 15.97 | | State = *"Dormant"*, Type = *All*, Style = *All*, Area of Mass = *All*, Date = *All*, Elevation = *All*, Rainfall = *All* |
| 3. When area of mass (m²) goes up 137.08, the average of causalities increases by 1.53 | | State = *"Stabilized"*, Type = *All*, Style = *All*, Area of Mass = *All*, Date = *All*, Elevation = *All*, Rainfall = *All* |
| 4. When rainfall (mm) goes up 29.29, the average of causalities increases by 0.49 | | State = *"Stabilized"*, Type = *All*, Style = *All*, Area of Mass = *All*, Date = *All*, Elevation = *All*, Rainfall = *All* |

**Table 2.** *Cont.*

| AI Insight | AI-Based System Settings | Scenario |
|---|---|---|
| 5. When area of mass (m²) goes up 71.31, the average of casualties increases by 0.69 | 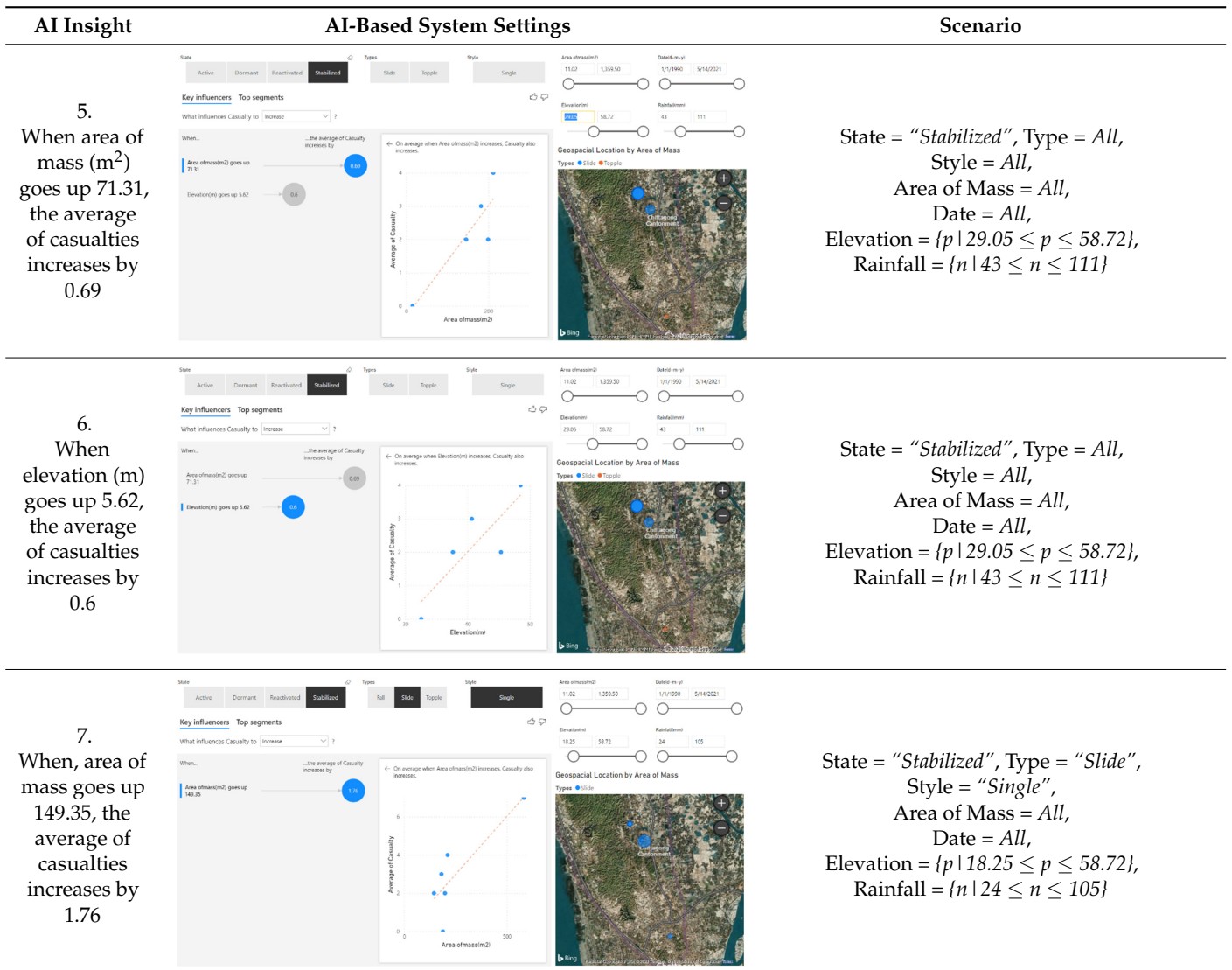 | State = *"Stabilized"*, Type = *All*, Style = *All*, Area of Mass = *All*, Date = *All*, Elevation = {$p \mid 29.05 \leq p \leq 58.72$}, Rainfall = {$n \mid 43 \leq n \leq 111$} |
| 6. When elevation (m) goes up 5.62, the average of casualties increases by 0.6 | | State = *"Stabilized"*, Type = *All*, Style = *All*, Area of Mass = *All*, Date = *All*, Elevation = {$p \mid 29.05 \leq p \leq 58.72$}, Rainfall = {$n \mid 43 \leq n \leq 111$} |
| 7. When, area of mass goes up 149.35, the average of casualties increases by 1.76 | | State = *"Stabilized"*, Type = *"Slide"*, Style = *"Single"*, Area of Mass = *All*, Date = *All*, Elevation = {$p \mid 18.25 \leq p \leq 58.72$}, Rainfall = {$n \mid 24 \leq n \leq 105$} |

The above scenario conditions can be located in the scenario column of Table 2.

Once the above scenario was configured using the software interface (as shown previously in Figure 8), the AI insight dynamically executed the regression analysis and described the following insights into plain English:

*When Area of Mass (m²) goes up 71.31, the average of Casualty increases by 0.69*

*When Elevation (m) goes up 5.62, the average of Casualty increases by 0.6*

In other words, for the selected scenario, casualties are positively correlated with both elevation and area of mass. The system dynamically calculated the coefficients of the positive correlation as soon as the user configured the scenario. Hence, the user of the system does not need to know the complexity of ML algorithms, and the user does not need to understand when to use linear regression and when to use logistic regression. The proposed interactive system executes the right regression depending on the configured scenario of the user. A strategic decision maker can obtain the AI insight in plain English and make appropriate decisions based on the AI insight.

The purpose of this section is not just to generate an exhaustive list of AI insights for all $1.054 \times 10^{41}$ possible scenarios. The rest of the Table 2 demonstrates some other AI insights generated by 4 other scenarios to demonstrate the applicability of the system.

Both Figures 9 and 10 show insights generated through decomposition analysis. Firstly, in Figure 9, a user selected the entire range of data using the option box and sliders at the top of the figure. Then, the user selected "High Value" (as shown previously in Figure 7) to find out what caused the highest number of casualties. Immediately after selection, the system showed the user that when type is "slide" casualty is highest. The system also provides visual cues to the user showing type = slide caused 130 casualties out of total 158 casualties. Hence, the user can confidently perform root cause analysis without any knowledge of underlying statistical methods. Furthermore, the user can select "High Value" again (as shown previously in Figure 7), and find out that when rainfall is 88, the number of casualties is at its peak (i.e., Figure 9 shows when rainfall is 88, there were 98 casualties). Similarly, the user can continue drilling down into further root causes to find out all of the features and the corresponding feature values that caused the highest number of recorded casualties. Following from a condition like "type is slide" and "rainfall is 88", Figure 9 shows the other feature conditions that caused the highest number of casualties, namely: "state is dormant", "area of mass is 1269.5", and "elevation is 46.07". Therefore, Figure 9 shows an interactive tool for discovering hidden insights into what caused the highest number of casualties.

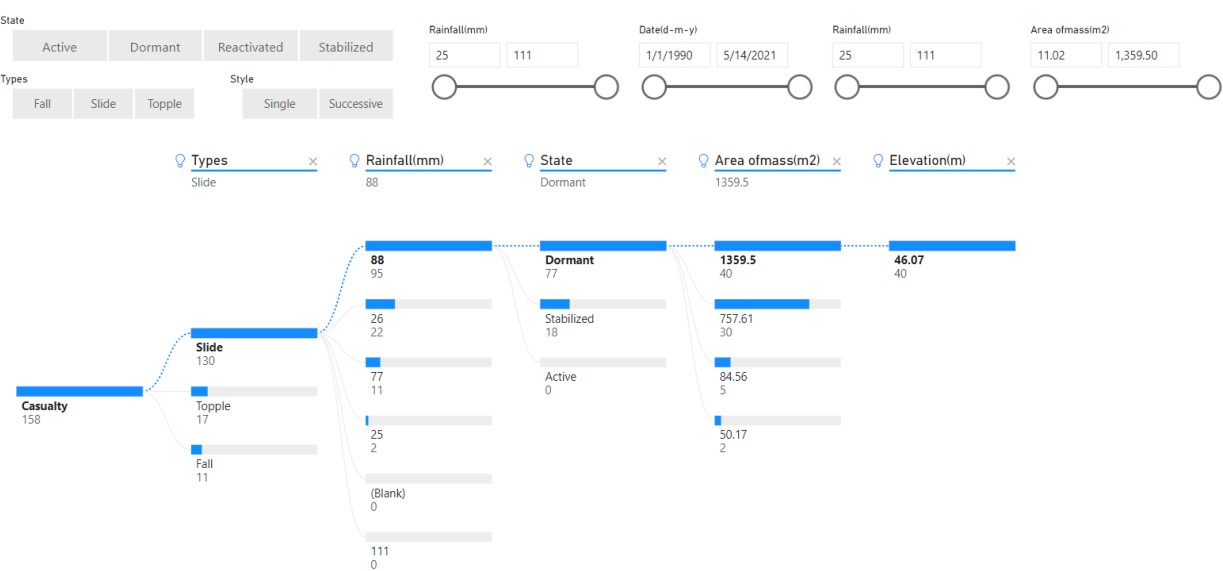

**Figure 9.** Decomposition analysis showing what causes the most casualties.

Using the decomposition tree visualization, the user can also find out what causes the lowest number of casualties and to find out how a particular feature effects the number of casualties. For example, Figure 10 shows what caused the highest casualties when types= "Topple". As depicted in Figure 9, the highest number of casualties (for types= "Topple") was found to be state = active (i.e., most important factor) and area of mass= 427.04 (m$^2$) (i.e., second most important factor), elevation = 31.66 (m) (i.e., third most important factor), and Rainfall = 55 (mm) (i.e., least important factor).

**Figure 10.** Decompression analysis showing what caused the highest casualties when types= "Topple".

It is crucial to highlight the fact that the proposed system is robust enough to provide critical insights from the underlying data on any number of scenarios as shown in Table 2, Figure 9, and Figure 10 using regression analysis and decompression analysis.

## 4. Discussion

Since all of the existing studies in landslide research do not support mobile app-based AI insight [1–28], it is not possible for a strategic decision maker to obtain instant insights if he or she is only equipped with mobile phone. In this study, we have deployed the proposed solution in desktop, tablet, and even mobile environments since the strategic decision maker can be eager to find out AI-based insights when they are remotely located at a possible landslide incident. As shown in Figure 11a, the AI-based auto-regression was executed on a Samsung Note 10 mobile phone. Figure 11b shows the decomposition analysis on the user's selected scenario was executed in a mobile environment as well. Figure 11c demonstrates the solution deployed through an iOS App on an Apple iPad 9th generation, running iOS version 15.1. Figure 11d showcases the deployed Android app running on a Samsung Galaxy Tab A7, running Android 11.

To test, assess, and evaluate the proposed AI-based landslide analysis system, the fully deployed solutions were given to 12 landslide researchers, disaster strategists, and town planners. The users were primarily located in the following area using their GPS-enabled devices for obtaining location-based insights using the proposed solution:

- Colony para, the University of Chittagong
- Motijharna, Chittagong City
- Matiranga, Rangamati

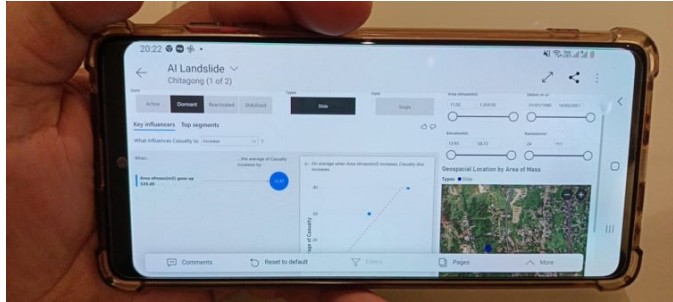

(**a**) Regression analysis on a Samsung Note 10 mobile device

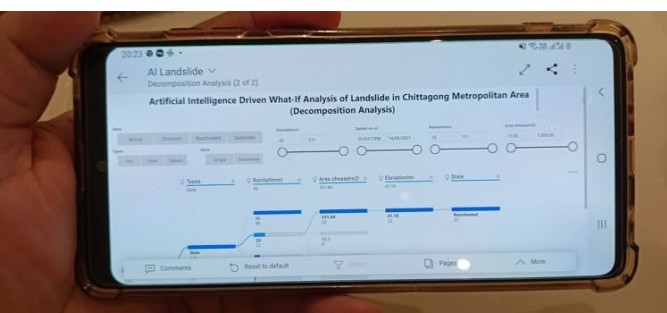

(**b**) Decompression tree analysis on a Samsung Note 10 mobile device

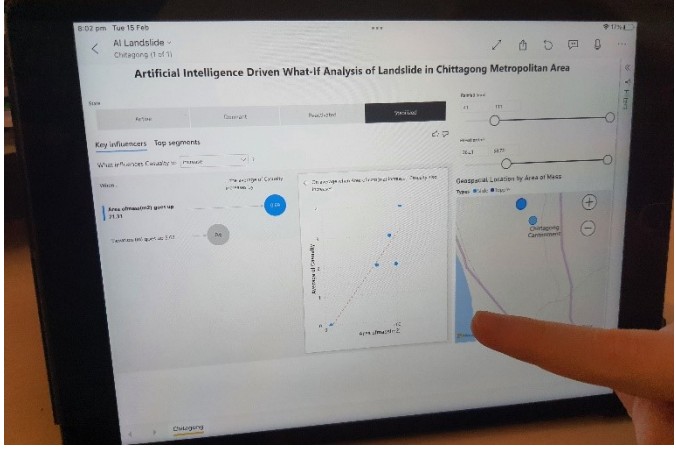

(**c**) Linear regression on an Apple iPad 9th Generation (iOS 15.1)

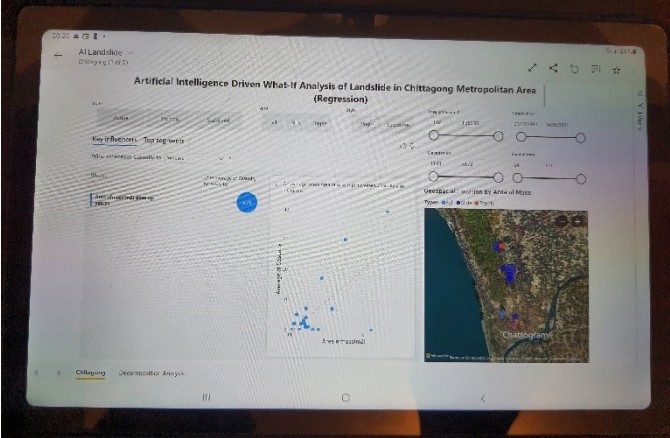

(**d**) Linear regression on a Samsung Galaxy Tab A7 (Android 11)

**Figure 11.** The proposed system running on mobile devices and providing AI-based insights on CMA landslides.

Table 3 shows the platform and user details for these tests and evaluations. As seen from Table 3, the proposed solution was tested on a wide range of devices, including both mobile devices and tablets. Since strategic decision makers often make their decisions on the site of the landslide or away from offices, they need mobile solutions deployed on tablets and mobile devices through iOS or Android apps. After the completion of the test and evaluation, detailed feedback regarding usability and appropriateness of the deployed solutions were obtained via Microsoft Form-based questionnaires (i.e., office 365 cloud-hosted). A total of 11 out of the 12 users (i.e., 91.67%) found the solution easy to use, effective, and self-explanatory. However, one user preferred using the solution in his desktop computer through the cloud-based interface.

**Table 3.** Details for deployment platforms and users.

| Number of Users | Device Name | OS Version |
| --- | --- | --- |
| 2 | Samsung Note 10 Lite (Mobile) | Android 11 |
| 1 | Samsung Note 10 Lite (Mobile) | Android 12 |
| 2 | Samsung Galaxy Tab A7 (Tablet) | Android 11 |
| 2 | iPhone 13 (Mobile) | iOS 15 |
| 1 | iPhone 12 (Mobile) | iOS 14 |
| 2 | iPad 9th Generation (Tablet) | iOS 15.2 |
| 2 | iPad Mini 6 (Tablet) | iOS 15 |

Hence, this mobile-based AI insight system provides a robust and innovative solution for the strategic decision maker who does not need to depend on a data scientist to conduct data modelling to obtain valuable insight. By interacting with the proposed system, a strategic decision maker can harness powerful ML algorithms automatically and obtain useful insights.

The process described within this paper is applicable for all different types of data on different types of scenarios to answer several different types of research questions. For example, this methodology was applied to obtain AI-driven insights on tornado-related casualties in Bangladesh [29]. Similarly, the study in [30] utilized this methodology in critically analyzing Australian cyclones. Moreover, this method could also be used to monitor disasters from any global location as demonstrated in [31,32] by analyzing live social media data. As shown in [31], AI-driven disaster intelligence solutions could be up to 97% accurate.

As it becomes apparent from these recent publications [29–33], it is first required that the dataset be cleansed and transformed. Pre-processing the available dataset with appropriate data cleansing and transformation is the key to obtaining better AI-driven insights on the casualties. Then, the Microsoft Power BI's key influencers visualization is used to analyze the outcome variable (e.g., casualties) with respect to a list of available "explain by" variables (e.g., elevation, rainfall, area of mass, longitude, latitude, number of injuries, style, types, etc.). The detailed process of using Microsoft Power BI's key influencers visualization is explained at https://learn.microsoft.com/en-us/power-bi/visuals/power-bi-visualization-influencers?tabs=powerbi-desktop, accessed on 20 December 2023. The machine learning (ML)-based feature analysis (e.g., linear regression or logistic regression) depends on the availability of many feature attributes for understanding their correlations to the outcome variable. In this study, casualty was deemed as an outcome variable, since strategic decision makers are always keen on saving precious lives resulting from landslides. Within our dataset, we only had few available features to analyze (e.g., latitude, longitude, elevation, area of mass, rainfall, etc.). After applying our innovative method, our solution found a positive correlation of casualty with area of mass (as shown in Figure 5, Row 1 of Table 2, Row 2 of Table 2, Row 3 of Table 2, Row 4 of Table 2, Row 5 of Table 2, Row 6 of Table 2, Row 7 of Table 2), rainfall (as shown in Row 3 of Table 2, Row 4 of Table 2), and elevation (as shown in Row 5 of Table 2, Row 6 of Table 2). Even though we utilized all of the available features present within our dataset to obtain relationships with the observed variable (i.e., casualty), we considered appropriate data cleansing prior to the automated ML process. As a result of the cleansing process, elevation and area of mass turned out to be a decimal data type and rainfall turned out to be integer data types.

## 5. User Notes

The ML-based knowledge discovery solution presented in this study was implemented using Microsoft Power BI, which is freely available for download from https://app.powerbi.com/, accessed on 20 December 2023. The user can download the complete source files (.pbix), along with the CMA landslide data (.csv) files from the author's GitHub site (i.e., https://github.com/DrSufi/landslide, accessed on 20 December 2023). After downloading and opening the entire solution using MS Power BI Desktop, the user can host the solution on either the Microsoft Cloud or within a local network to make it available to other researchers or strategic planners.

The typical users of this system are strategic disaster planners, disaster risk assessors, policymakers, and disaster strategists who are concerned with landslides or landfalls and their subtle impact on society, groups, and locations. This system would allow users to understand the characteristics of global events in a particular area since it provides useful guidance for policy implementation.

## 6. Conclusions

This paper provides a detailed methodological framework for generating AI-based insights on landslides in the CMA. This experimentation was performed on a limited dataset containing only 57 records. Sadly, there were several limitations due to the relatively small dataset in terms of empty values within date and rainfall attributes. As is evident from Table 1, the date attribute has 34 empty values (i.e., 40% valid and 60% empty values) and rainfall has 18 empty values (i.e., 68% valid and 32% empty).

AI-based automated insight generation processes as depicted in this research are often referred to as data-driven insights. For data-driven insights, having a robust and complete set of data is often a mandate. In case the data suffers from irregular/missing values (or any other data quality issues hampering the overall quality of the dataset) then several pre-processing techniques (e.g., StandardScaler, MinMaxScaler, StandardScaler, OneHotEncoder, etc.) could enhance the performance of data-driven insight solutions. Despite these limitations on available information, the AI-based techniques like automated regressions (both linear and logistic) as well as a decomposition algorithm successfully derived useful insights for the strategic decision maker.

In the future, we will endeavor to work with more records of landslides outside of the CMA region. Using these large-scale records, we hope to deploy more sophisticated AI-based techniques like convolution neural network (CNN)-based deep learning to generate useful insights (since our recent study in [30–32] has demonstrated that applying CNN on disaster monitoring harnesses better results). Other than CNN, we also want to use sophisticated AI-based techniques as demonstrated in our recent and past studies [29–34].

**Author Contributions:** Conceptualization, E.A. and F.S.; methodology, F.S. and E.A.; A.R.M.T.I.; validation, F.S., E.A. and A.R.M.T.I.; resources, E.A.; data curation, F.S.; writing—original draft preparation, E.A. and F.S.; writing—review and editing, E.A., A.R.M.T.I. and F.S.; visualization, F.S.; supervision, E.A. All authors have read and agreed to the published version of the manuscript.

**Funding:** This research received no external funding.

**Institutional Review Board Statement:** Not applicable.

**Informed Consent Statement:** Not applicable.

**Data Availability Statement:** Data are available upon request.

**Acknowledgments:** Support received for data collection from the Bangladesh from Disaster Action and Development Organisation (DADO) is highly appreciated. Funding support for article processing charges (APCs) from the Rabdan Academy is greatly appreciated.

**Conflicts of Interest:** The authors declare no conflict of interest.

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
