# Peer review of "A Scenario-Based Case Study: Using AI to Analyze Casualties from Landslides in Chittagong Metropolitan Area, Bangladesh"

_sustainability, doi:10.3390/su15054647_

Round 1

Reviewer 1 Report

Comments:

Comment 1:       1.054X1041 to be corrected [Line 18]

Comment 2:       No justification for choosing ANN over other statistical methods or any other methods
                                mentioned in the document [Line 98 – 103]

Comment 3:       Census data up to 2011 is mentioned latest Census data or forecast is not considered.
[Line 145 – 147]

Comment 4:       Figures can be mentioned as (a), (b), (c), (d) and caption for each to be inserted.
[Line 163 – 164]

Comment 5:       For table 1, in page no 9 heading for the columns should be provided. [Line 281]

Comment 6:       Population density not considered among the different attributes. May have correlation with dataset. [Line 296 – 297]

Comment 7:       Figure no 11, 12, 13 and 14 are repetitive in nature. Can be provided in annexure or as collage image. [Line 425 – 426]

Comment 8:       Link for the mentioned (.pbix) and (.csv) files are not provided in the document [Line 448 – 451]

Comment 9:       Indentation is not maintained uniformly throughout the document.

Comment 10:     With thorough reading typographical errors to be reduced. i.e., ‘reason’ [Line 131] and ‘date’ [Line 172]

Author Response

Thanks for constructive feedback. We have responded to all comments.

Reviewer 2 Report

The paper "A Scenario-based Case Study: AI to analyse casualties from 2 landslides in Chittagong Metropolitan Area, Bangladesh" uses an AI approach to forecasting landslide impacts. The Materials & Methods appear sound, the results explore a mulititude of possible scenarios (1.054 x 10^41) during processing, including the dynamic link to existing databases, and the relatively easy use by practioners who receive effective information. 

The struture of the paper is fine even though there is much information for the reader to digest, however all components of the approach appear to be discussed (summarised in Fig 6). This is a another innovative approach to AI and I encourage the authors to continue this research.

Author Response

Thanks for constructive feedback. We have responded to all comments

Reviewer 3 Report

I recommend the manuscript revise follow:

1. Authors is necessary to modify the text form (From p. 8~ ).
2. Tables were difficult to understand, I recommended that authors needs overall revision of table.
3. In Equations 9 to 15, seems to unnecessary. I recommended it explain in manuscript.
4. What is the meaning of lines 345-351 ?
Authors add description or/and insert into figure.
5. I recommended that Figures 11-14 need to be incorporated.
6. I found the typo errors.
7. Overall, the manuscript was a lack of trust in data from AI.
8. Why did you need to apply AI ?
Authors find it and write it in the manuscript.
9. How to verify your proposed model? And the level of skill in question should be described.
10. I recommended that delete unnecessary UI descriptions. If so, explain how the basic variables used in the UI were calculated.

I decided major revision.

Author Response

Thanks for your constructive feedback. We have responded to all comments

Round 2

Reviewer 1 Report

Comment 1:       [Reference to comment 3], The updated value of population from 0.9 million to 13 million is mentioned as 6 times increase [Line 150-153]

Comment 2:       [Reference to comment 4],  Not corrected. Figure 2 can be modified in same way as modification done for figure 11. [Line 166]

All the other changes made based on other comments are satisfactory and accepted.

Author Response

Please find the attached response notes addressing your comments

Reviewer 3 Report

I recommend the accept for this manuscript.

Author Response

Please find attached response letter
